# Obesity causes selective and long-lasting desensitization of AgRP neurons to dietary fat

Lisa R Beutler[1], Timothy V Corpuz[2], Jamie S Ahn[2], Seher Kosar[2], Weimin Song[3], Yiming Chen[4,5], Zachary A Knight[2,4,5,6]*

[1]UCSF Department of Medicine, San Francisco, United States; [2]Howard Hughes Medical Institute, Chevy Chase, United States; [3]Northwestern University Feinberg School of Medicine, Comprehensive Metabolic Core, Chicago, United States; [4]UCSF Department of Physiology, San Francisco, United States; [5]UCSF Neuroscience Graduate Program, San Francisco, United States; [6]Kavli Institute for Fundamental Neuroscience, San Francisco, United States

**Abstract** Body weight is regulated by interoceptive neural circuits that track energy need, but how the activity of these circuits is altered in obesity remains poorly understood. Here we describe the in vivo dynamics of hunger-promoting AgRP neurons during the development of diet-induced obesity in mice. We show that high-fat diet attenuates the response of AgRP neurons to an array of nutritionally-relevant stimuli including food cues, intragastric nutrients, cholecystokinin and ghrelin. These alterations are specific to dietary fat but not carbohydrate or protein. Subsequent weight loss restores the responsiveness of AgRP neurons to exterosensory cues but fails to rescue their sensitivity to gastrointestinal hormones or nutrients. These findings reveal that obesity triggers broad dysregulation of hypothalamic hunger neurons that is incompletely reversed by weight loss and may contribute to the difficulty of maintaining a reduced weight.

*For correspondence: zachary.knight@ucsf.edu

Competing interests: The authors declare that no competing interests exist.

## Introduction

Body weight is regulated by a physiologic system that balances energy intake and expenditure over the long term. Key components of this system are specialized neurons in the hypothalamus that monitor internal state, but how obesity alters these circuits remains poorly understood.

Among the cell types involved in body weight regulation, AgRP neurons are particularly important. These neurons are located in the arcuate nucleus of the hypothalamus and are activated by food deprivation (*Hahn et al., 1998*; *Mandelblat-Cerf et al., 2015*; *Takahashi and Cone, 2005*; *van den Top et al., 2004*). Stimulation of AgRP neurons drives voracious feeding and recapitulates the motivational, affective and sensory hallmarks of hunger (*Aponte et al., 2011*; *Chen et al., 2016*; *Krashes et al., 2011*; *Livneh et al., 2017*). Conversely, silencing or ablating AgRP neurons causes aphagia (*Gropp et al., 2005*; *Krashes et al., 2011*; *Luquet et al., 2005*). Thus AgRP neurons link the need for energy to the desire to eat.

AgRP neurons are regulated by layers of nutritional signals that operate on different time scales. Leptin, which is secreted from adipose tissue in proportion to body fat reserves, influences AgRP neuron activity over the long term (*Beutler et al., 2017*; *Coleman, 1978*; *Cowley et al., 2001*; *Pinto et al., 2004*; *Seeley et al., 1997*; *van den Top et al., 2004*; *Vong et al., 2011*; *Zhang et al., 1994*). On shorter timescales, AgRP neurons receive two rapid signals that report on impending or recent food consumption. The first is triggered by the sensory detection of food in the environment and functions to inhibit AgRP neurons within seconds (*Betley et al., 2015*; *Chen et al., 2015*; *Mandelblat-Cerf et al., 2015*). The magnitude of this rapid sensory inhibition predicts the quantity of

imminent food intake. Infusion of nutrients directly into the stomach bypasses this sensory response and reveals additional signals that arise from the gastrointestinal (GI) tract (*Beutler et al., 2017*; *Su et al., 2017*). These are trigged by the detection of calories or the activation of mechanoreceptors (*Bai et al., 2019*) and inhibit AgRP neurons over minutes as nutrients are infused. Thus AgRP neurons integrate multiple signals, arising from different tissues, in order to estimate the body's energy needs.

A critical unanswered question regards how obesity alters AgRP neuron activity in response to these diverse signals. It is thought that diet-induced obesity renders AgRP neurons resistant to the effects of leptin, based in part on the fact that exogenous leptin fails to reduce food intake or induce pSTAT3 in obese animals and does not inhibit AgRP neurons in ex vivo preparations from obese animals (*Bates et al., 2003*; *Baver et al., 2014*; *Enriori et al., 2007*; *Knight et al., 2010*; *Wei et al., 2015*). In contrast, nothing is known about how diet-induced obesity alters the response of AgRP neurons to sensory cues signaling food availability or their modulation by gastrointestinal nutrients during a meal. These aspects of AgRP neuron regulation cannot be probed using ex vivo methods, such as immunohistochemistry or slice physiology, due to their rapid timescale, inhibitory nature, and because the pathways for gut-brain communication are disrupted in a brain slice.

To address this basic question, we set out to monitor and manipulate the activity of AgRP neurons in vivo in mice during the development of obesity induced by a high-fat diet (HFD). We have further tracked how these responses change during weight loss. We show that diet-induced obesity attenuates neural and behavioral responses to multiple nutritional stimuli including food presentation, hormone administration and intragastric (IG) nutrient delivery. These changes are nutrient-specific, with blunted AgRP neuron responses to fat but not glucose or protein. Moreover, while the AgRP neuron response to the sensory detection of food is rescued with the onset of weight loss, neural responses to gastrointestinal stimuli are not. This combination of changes represents a neural correlate for the long-lasting effects of obesity on the energy homeostasis system.

## Results

### Diet-induced obesity attenuates the AgRP neuron response to the sensory detection of food

We set out to characterize how the regulation of AgRP neurons is modulated by diet-induced obesity and subsequent weight loss. We generated mice equipped for fiber photometry recordings from AgRP neurons (*Beutler et al., 2017*; *Chen et al., 2015*) and then tested these animals at baseline, after 6 weeks of ad libitum consumption of HFD (60% kcal from fat), and then again after 4 weeks of ad libitum chow consumption (*Figure 1A*). Exposure to HFD produced reliable weight gain, which was partially reversed after animals were returned to chow (*Figure 1B,C*). As an additional control, we also recorded from lean mice that were maintained in parallel on standard rodent chow (13% kcal from fat) for the duration of the study.

We measured the response of AgRP neurons to the sensory detection of food along this time course. Mice were fasted overnight and then presented with a pellet of chow or high-fat diet and photometry responses recorded. As previously reported (*Betley et al., 2015*; *Chen et al., 2015*; *Mandelblat-Cerf et al., 2015*), food presentation rapidly inhibited AgRP neurons of lean mice at baseline (*Figure 1D,G*). After six weeks on HFD, obese animals showed markedly attenuated responses to presentation of both chow and HFD (*Figure 1E,F,H,I,J,K,M,N*). Of note, the amount of weight gain in individual mice did not correlate significantly with magnitude of the reduction in AgRP neuron response to food presentation (*Figure 1—figure supplement 1*), indicating that this impairment is not solely due to increased body weight.

Diet-induced obese (DIO) mice were then switched back to chow and tested again after four weeks of weight loss (*Figure 1A,B*), which partially rescued the response of AgRP neurons to chow (*Figure 1E,F,L*) but not HFD presentation (*Figure 1H,I,O*). In contrast, control mice maintained on chow throughout this time course exhibited relatively stable neuronal inhibition upon presentation of both foods. Thus, diet-induced obesity reversibly blunts the rapid inhibition of AgRP neurons in response to exterosensory cues.

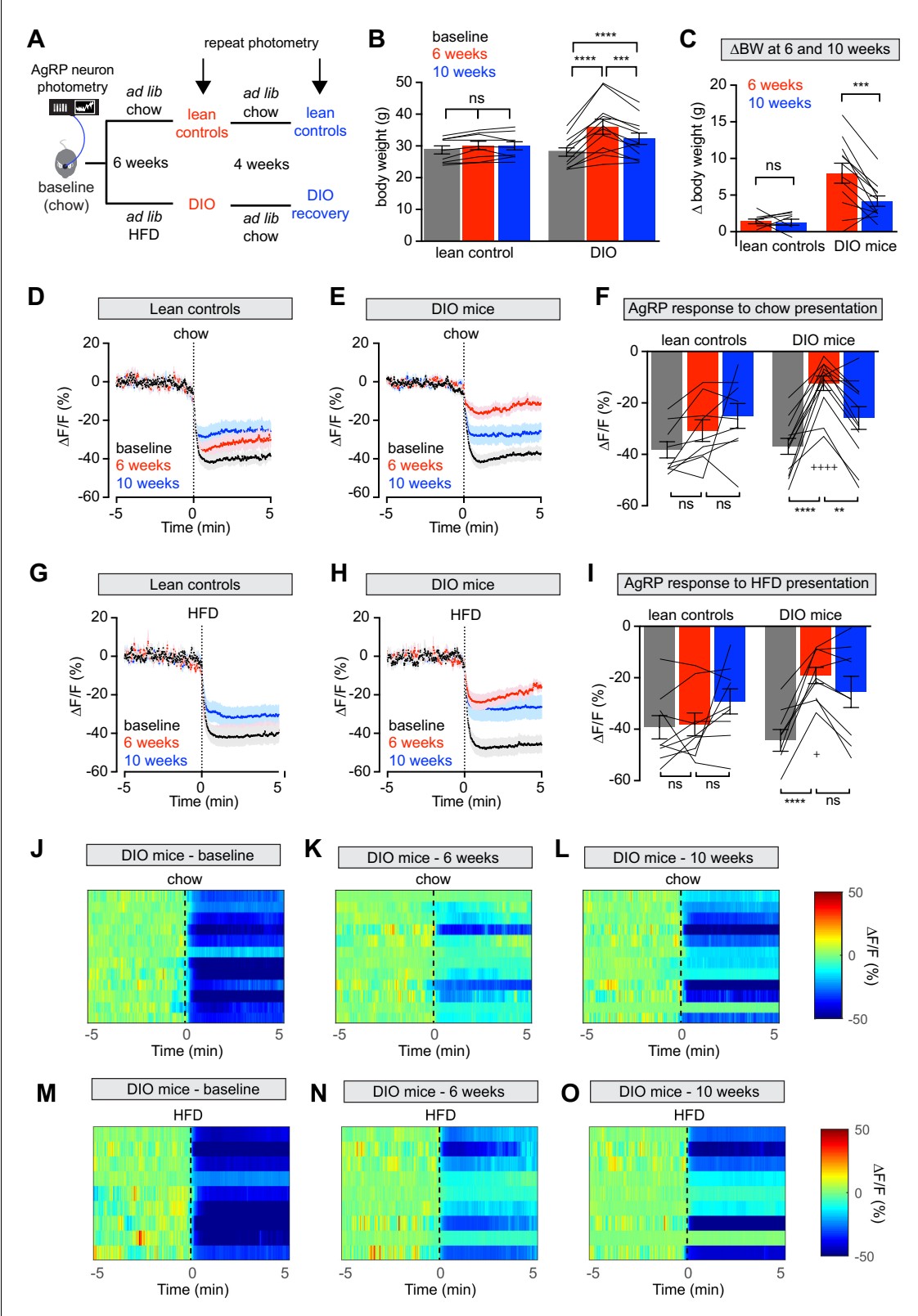

**Figure 1.** Diet-induced obesity (DIO) reversibly attenuates AgRP neuron inhibition in response to food presentation. (A) Schematic for the photometry experiments performed in this study. Mice were divided into two cohorts, one of which received chow throughout (lean controls) and one of which was challenged with high-fat diet (HFD) for 6 weeks and then returned to chow for 4 weeks (DIO mice). Baseline recordings were taken from AgRP neurons on day 0, then repeated after 6 weeks on either HFD or chow, and then repeated 4 weeks later after all DIO mice had been returned to chow. (B) Body

*Figure 1 continued on next page*

*Figure 1 continued*

weights in lean control (left) and DIO (right) mice at baseline, after 6 weeks of chow or HFD, and after an additional 4 weeks of chow (n = 9–12 mice per group). (C) Change in body weight from baseline in mice from (B) (D and E) Calcium signal from AgRP neurons in fasted control (D) and DIO (E) mice presented with chow at baseline (black), after 6 weeks of chow or HFD (red), and after an additional 4 weeks of chow (blue). (n = 9–12 mice per group). (F) Quantification of ΔF/F from (D) and (E) for 5 min after chow presentation. (G and H) Calcium signal from AgRP neurons in fasted control (G) and DIO (H) mice presented with HFD at baseline (black), after 6 weeks of chow or HFD (red), and after an additional 4 weeks of chow (blue). (n = 9 mice per group). (I) Quantification of ΔF/F from (G) and (H) for 5 min after HFD presentation. (J–L) Peri-stimulus heatmaps depicting ΔF/F of AgRP neurons in individual DIO mice (from E) following presentation of chow at baseline (J), 6 weeks (K) and after 4 weeks recovery (L). (M–O) Peri-stimulus heatmaps depicting ΔF/F of AgRP neurons in individual DIO mice (from H) following presentation of HFD at baseline (M), 6 weeks (N) and after 4 weeks recovery (O). *p<0.05, **p<0.01, ***p<0.001 and ****p<0.001 as indicated. +p<0.05, ++++p<0.0001 compared to lean control at the 6 week timepoint. There was no significant difference between lean control and DIO groups at baseline. (D,E,G,H) Traces represent mean ± SEM (B,C,F,I) Lines denote individual mice. Error bars represent mean ± SEM.

The online version of this article includes the following figure supplement(s) for figure 1:

**Figure supplement 1.** Magnitude of DIO-induced changes in AgRP neuron dynamics are not correlated with amount of weight gain.

## Diet-induced obesity decreases food consumption after fasting

The rapid inhibition of AgRP neurons by food presentation is thought to be a prediction of impending food consumption (*Beutler et al., 2017*; *Chen et al., 2015*). We therefore tested whether these HFD-induced changes in AgRP neuron dynamics are correlated with changes in subsequent food intake. Mice were fasted overnight and then given access to food and consumption over 30 min was measured at the time points in *Figure 1*. We found that DIO animals consumed significantly fewer calories when presented with chow after an overnight fast compared to lean controls (*Figure 2A and B*), as others have observed (*Briggs et al., 2011*; *Ueno et al., 2007*). This reduction in caloric intake was not due solely to the lower palatability of chow, because DIO mice that were fasted and then allowed to re-feed with HFD also showed a reduction in consumption of HFD when compared either to lean controls (*Figure 2C and D*) or their own HFD consumption prior to obesity (*Figure 3E*). This reduction in HFD consumption by DIO animals was also not due to sensory-specific satiety or a learned aversion, because when fasted DIO animals were presented with chocolate, a

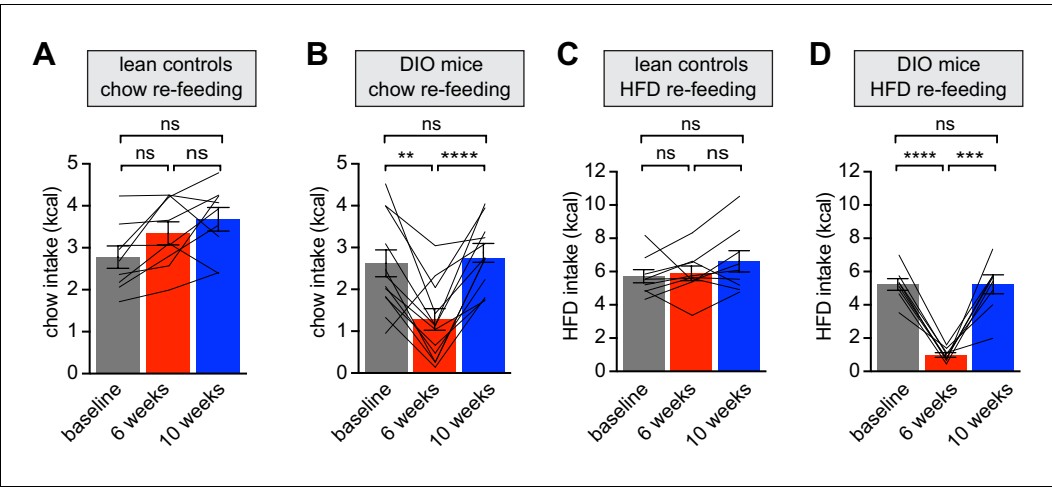

**Figure 2.** Diet-induced obesity reversibly attenuates fasting induced re-feeding. (A–D) Animals were maintained on either a chow diet throughout the entire experiment (lean controls, **A and C**) or exposed to high-fat diet for 6 weeks before being returned to chow diet for an additional 4 weeks (DIO mice, **B and D**). At 0, 6 and 10 weeks, animals were fasted overnight, and then re-fed for 30 min with chow (**A and B**) or HFD (**C and D**) and caloric intake was recorded (n = 8–12 mice per group). **p<0.01, ***p<0.001, ****p<0.0001 as indicated. Lines denote individual mice. Error bars represent mean ± SEM.

The online version of this article includes the following figure supplement(s) for figure 2:

**Figure supplement 1.** Hyperleptinemia in diet-induced obesity correlates with suppression of fast re-feeding.

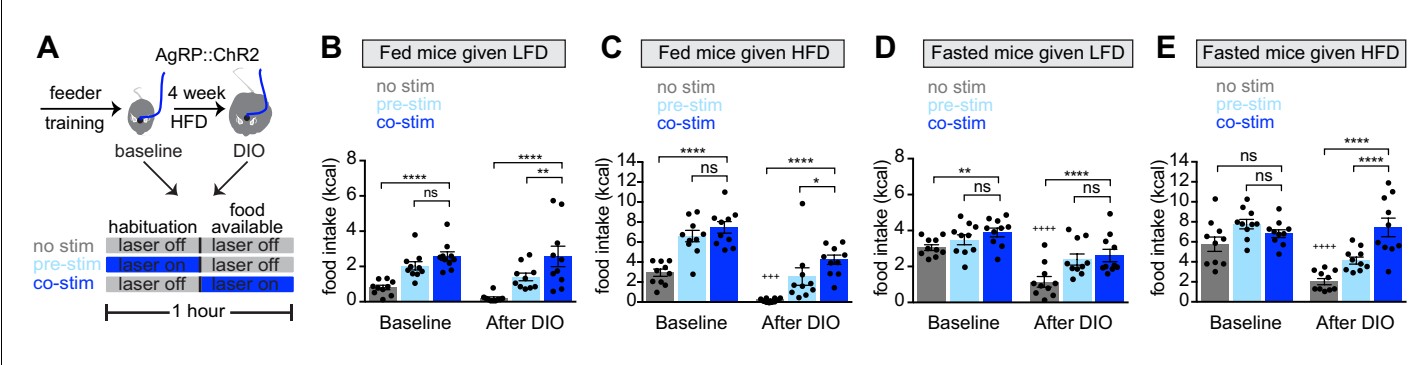

**Figure 3.** Obesity alters the ability of AgRP neurons to drive food consumption. (**A**) Schematic for optogenetic experiments. Animals equipped for optogenetic stimulation of AgRP neurons were trained to retrieve pellets from a feeder then exposed to three optical stimulation (stim) conditions on different days. All sessions consisted of 30 min habituation without food availability followed by 30 min of food availability. During no stim sessions, the laser remained off throughout this duration. During pre-stim sessions, laser - stimulation occurred during habituation only. During concurrent-stim (co-stim) sessions, laser stimulation occurred during food availability only. Each animal was tested using either low-fat diet (LFD, grain pellets) or high-fat diet (HFD) as the available food and in the fed and fasted states . Each experiment was performed on the same animals at baseline and after 4 weeks of ad libitum HFD intake for a total of 12 sessions. (**B–C**) Calories consumed by fed mice as LFD (**B**) or HFD (**C**) at baseline and after DIO induction under no stim (gray), pre-stim (light blue), and co-stim (dark blue) conditions. (**D–E**) Calories consumed by fasted mice as LFD (**D**) or HFD (**E**) at baseline and after DIO induction under no stim (gray), pre-stim (light blue), and co-stim (dark blue) conditions. n = 10 animals; *p<0.05, **p<0.01, ****p<0.0001 as indicated; +++ p<0.001, ++++ p<0.0001 DIO no stim compared to baseline no stim. (**B–E**) • denotes individual mice. Error bars represent mean ± SEM.

novel palatable food, their food intake remained significantly attenuated (control-chocolate 1.36 kcal vs DIO-chocolate 0.36 kcal, p<0.001).

Animals were returned to chow for four weeks, which resulted in moderate weight loss (*Figure 1B*). However this dietary switch completely restored the consumption of chow and HFD in formerly DIO mice following an overnight fast (*Figure 2B and D*). Strikingly, these behavioral responses to weight gain and loss mirrored the changes in AgRP neuron responses to food during this time course (*Figure 1*). This reveals that HFD exposure inhibits fasting-induced hyperphagia, which is then rapidly restored following weight loss, and that the anticipatory dynamics of AgRP neurons track these behavioral fluctuations.

To further investigate potential metabolic mechanisms underlying blunted fasting-induced hyperphagia in obese mice, we placed wild type mice on a HFD for four weeks while maintaining control mice on a normal chow diet. These DIO animals exhibited attenuated fasting-induced re-feeding when presented with chow or HFD despite limited weight gain over this short time course (*Figure 2—figure supplement 1A–D*). As expected, plasma leptin was significantly elevated in DIO animals, but fasting insulin levels were not elevated relative to lean controls (*Figure 2—figure supplement 1E and F*). Moreover, fasting-induced re-feeding was negatively correlated with leptin levels in DIO animals presented with HFD whereas insulin levels did not significantly correlate with food intake (*Figure 2—figure supplement 1G–J*). Thus, hyperleptinemia is correlated with decreased fasting-induced hyperphagia in mice early in DIO.

## Obesity alters the ability of AgRP neurons to drive food consumption

AgRP neuron activity is required for food deprivation to trigger compensatory re-feeding (*Denis et al., 2015*; *Liu et al., 2012*). We therefore investigated whether the decreased fasting-induced re-feeding we observe in obese mice might be due to a change in the ability of AgRP neurons to drive food intake.

We equipped mice for optogenetic stimulation of AgRP neurons and then tested them, before and after the development of obesity, using two complementary stimulation protocols. In the first protocol ('pre-stimulation'), AgRP neurons were stimulated for 30 min immediately prior to food availability, and then mice were given food and allowed to eat in the absence of continued stimulation (*Chen et al., 2016*). This protocol is designed to mimic the natural dynamics of AgRP neurons, in which these neurons are active in hungry mice but shut off immediately before food consumption.

In the second paradigm ('concurrent stimulation'), mice were given food and AgRP neurons were concurrently stimulated for 30 min while mice ate (*Figure 3A*). This stronger stimulation protocol has been shown to drive feeding behavior to a greater extent than natural hunger (*Burnett et al., 2019*).

We first tested lean animals at baseline. As expected, we found that concurrent or pre-stimulation of AgRP neurons caused fed mice to eat more low-fat pellets (*Figure 3B*, baseline) or HFD (*Figure 3C*, baseline), and the amount of food consumed was similar between the two protocols (*Figure 3B,C*, baseline). In the fasted state, pre-stimulation did not increase the consumption of either diet (*Figure 3D,E*, baseline), whereas concurrent stimulation caused a small increase in the consumption of low-fat pellets (*Figure 3D*, baseline) but not HFD (*Figure 3E*, baseline). This result is consistent with the notion that AgRP neurons are already highly active in fasted, lean mice, and therefore that optogenetic stimulation has little additional effect.

Animals were then placed on a HFD to induce weight gain and, four weeks later, the tests above were repeated. Consistent with our data from a separate cohort (*Figure 2*), we found that, in the absence of optogenetic stimulation, diet-induced obesity greatly reduced the amount of food the mice consumed during the behavioral tests. For example, diet-induced obesity blunted the hyperphagia in response to presentation of HFD (*Figure 3C*, no stim, baseline versus DIO) as well as the hyperphagia induced by fasting (*Figure 3D,E* no stim, baseline versus DIO). Importantly, we found that optogenetic stimulation of AgRP neurons could rescue this decreased feeding in DIO mice (*Figure 3B–E*, DIO). This suggests AgRP neurons are not fully activated by fasting in DIO animals, and that optogenetic stimulation supplies this missing activation.

In obese mice, we found that concurrent stimulation induced a bigger effect on food intake than pre-stimulation (*Figure 3B,C,E*, DIO), whereas in lean animals the effect of these two stimulation protocols was indistinguishable in every test performed (*Figure 3B–E*, baseline). One interpretation of this result is that diet-induced obesity reduces the sensitivity of the downstream circuitry to AgRP neuron activity, such that supraphysiologic stimulation (i.e. concurrent stimulation) is now required to fully drive feeding. This suggests that HFD causes dysregulation of both AgRP neurons and their downstream targets.

## Obesity selectively reduces the AgRP neuron response to intragastric fat

AgRP neurons are regulated not only by the sensory detection of food but also by ingested nutrients. This gastrointestinal regulation can be selectively elicited by infusing food directly into the stomach, which inhibits AgRP neurons in proportion to the number of calories infused but independent of the macronutrient composition of the infusate (*Beutler et al., 2017*; *Su et al., 2017*). We therefore set out to test whether obesity modulates this nutrient-mediated signaling from the gastrointestinal tract to the hypothalamus.

Animals prepared for AgRP neuron photometry recordings were additionally equipped for nutrient infusion via intragastric catheters (*Beutler et al., 2017*). We first tested these animals at baseline by infusing solutions of pure lipid, sugar or protein into the stomach over the course of 12 min (*Figure 4* and *Figure 4—figure supplement 1*). As previously reported, all three infusates induced progressive, potent, and durable inhibition of AgRP neuron activity in hungry animals (*Figure 4A,G* and *Figure 4—figure supplement 1A*; *Beutler et al., 2017*; *Su et al., 2017*). We then exposed these animals to HFD, or for the control group maintained them on chow, and repeated these infusions. Strikingly, AgRP neuron inhibition in response to lipid infusion was significantly decreased in DIO animals compared to their response at baseline (*Figure 4B–E*). In contrast the response of AgRP neurons to glucose and protein was not altered in obese mice (*Figure 4H–K* and *Figure 4—figure supplement 1B–D*), and control animals maintained on a low-fat diet showed no change in their response to any macronutrient (*Figure 4A,G* and *Figure 4—figure supplement 1A*). This indicates that obesity induced by HFD selectively desensitizes AgRP neurons to dietary fat.

To test whether weight loss reverses this desensitization, we returned animals to a chow diet for four weeks and then repeated the macronutrient infusions. Weight loss did not rescue the neural response to IG lipid in DIO mice, which remained significantly impaired relative to baseline measurements (*Figure 4B–F*), whereas control mice maintained a stable neural response to lipid throughout the experiment (*Figure 4A*). Thus, HFD-mediated obesity selectively attenuates lipid-induced

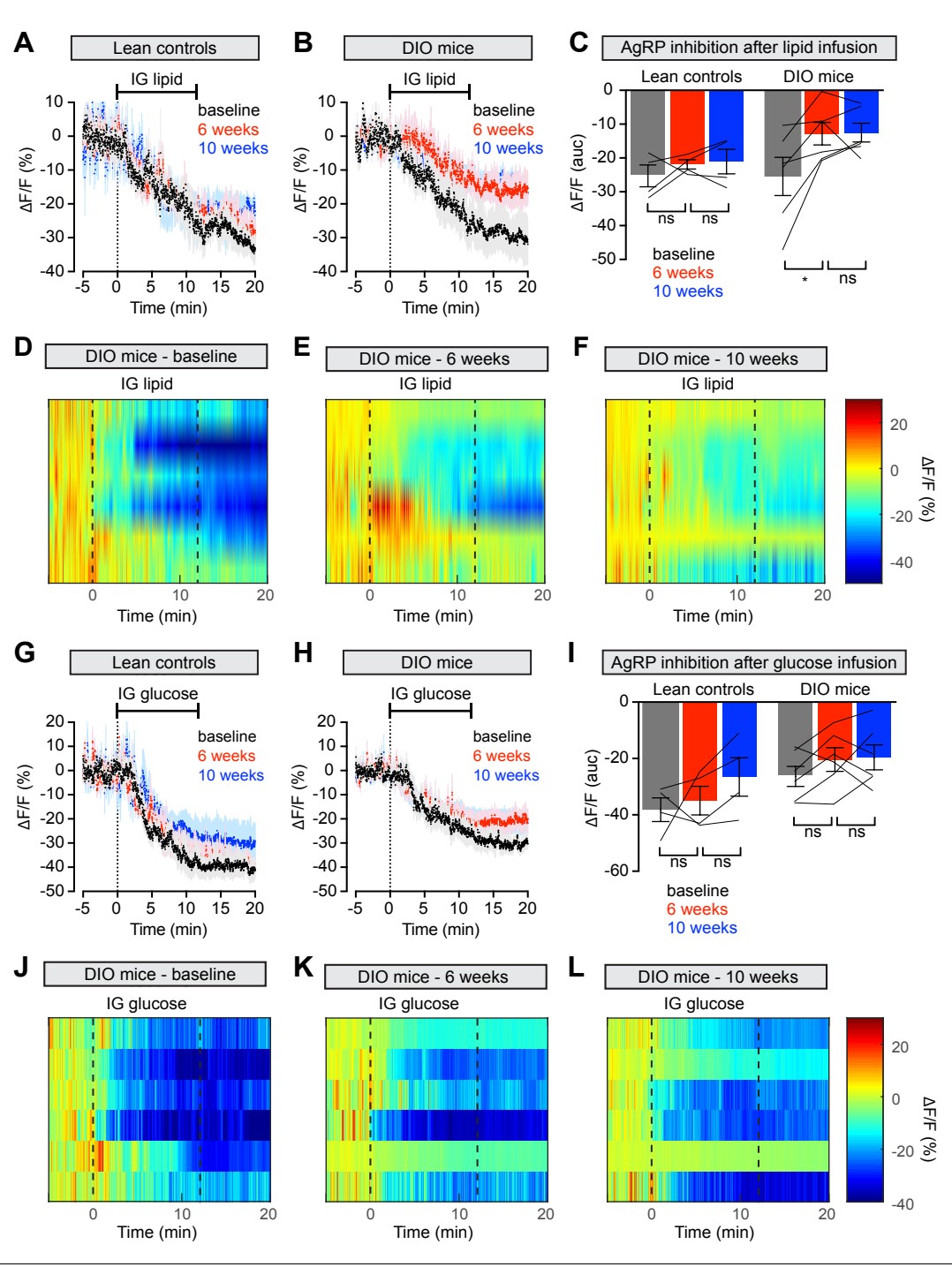

**Figure 4.** Diet-induced obesity causes selective and long-lasting attenuation of AgRP neuron inhibition by lipid. (A and B) Calcium signal from AgRP neurons in fasted control (A) and DIO (B) mice during intragastric infusion with intralipid at baseline (black), after 6 weeks of chow or HFD (red), and after an additional 4 weeks of chow (blue). (n = 4–6 mice per group) (C) Quantification of ΔF/F from (A) and (B) showing inhibition at the end of infusion. (D–F) Peri-infusion heatmaps depicting ΔF/F during photometry recording in individual DIO mice (from B) at baseline (D), after 6 weeks on HFD (E), and after 4 weeks recovery (F). (G and H) Calcium signal from AgRP neurons in fasted control (G) and DIO (H) mice during intragastric infusion with glucose at baseline (black), after 6 weeks of chow or HFD (red), and after an additional 4 weeks of chow (blue). (n = 4–6 mice per group) (I) Quantification of ΔF/F from (G) and (H) showing inhibition at the end of infusion. (J–L) Peri-infusion heatmaps depicting ΔF/F during photometry recording in individual DIO mice (from H) at baseline (J), after 6 weeks on HFD

*Figure 4 continued on next page*

*Figure 4 continued*

(**K**), and after 4 weeks recovery (**L**). *p<0.05 as indicated. There was no significant difference between lean control and DIO groups at baseline. (**A,B,G,H**) Traces represent mean ± SEM. (**C,I**) Lines denote individual mice. Error bars represent mean ± SEM.

The online version of this article includes the following figure supplement(s) for figure 4:

**Figure supplement 1.** Diet-induced obesity does not alter AgRP neuron response to intragastric infusion of peptides.

inhibition of AgRP neurons and modest weight loss is not sufficient to restore the effects of this macronutrient.

## Diet-induced obesity blunts AgRP neuron responses to cholecystokinin (CCK) and ghrelin

The mechanisms by which gastrointestinal nutrients inhibit AgRP neurons are poorly understood, but it has been shown that the hormone CCK is required for the effects of dietary fat on these cells (*Beutler et al., 2017*). We therefore investigated whether the inhibition of AgRP neurons by CCK might be altered in DIO animals.

Mice were fasted overnight, challenged with an intraperitoneal injection of CCK, and AgRP neuron responses recorded by photometry. At baseline CCK injection caused potent but transient inhibition of AgRP neuron activity, as previously reported (*Figure 5A*), and led to suppression of fasting-induced re-feeding (*Figure 5M*). Both the neuronal and behavioral responses to CCK were reduced following six weeks of HFD exposure (*Figure 5B–E,N*; *Figure 5—figure supplement 1A–D*) but unchanged in control animals maintained on a chow diet. As observed for chow presentation, the reduction in AgRP neuron dynamics did not correlate with weight gain, suggesting that this impairment is not solely due to increased weight (*Figure 5—figure supplement 1I*). Returning DIO mice to a chow diet for four weeks failed to rescue this CCK resistance (*Figure 5B–F*). Thus HFD causes long-lasting desensitization of AgRP neurons to the effects of CCK, which may explain why these neurons show persistent desensitization to dietary fat.

In addition to CCK, ghrelin is also known to regulate food intake via AgRP neurons (*Chen et al., 2015*; *Cowley et al., 2003*; *Hewson and Dickson, 2000*; *Kamegai et al., 2001*; *Nakazato et al., 2001*; *Tang-Christensen et al., 2004*; *van den Top et al., 2004*; *Wang et al., 2014*; *Zigman et al., 2005*) and obesity is thought to cause ghrelin resistance in AgRP neurons as measured by Fos staining and ex vivo recordings (*Briggs et al., 2010*; *Briggs et al., 2014*). To characterize this ghrelin resistance in vivo, we first confirmed that ghrelin injection activates AgRP neurons (*Figure 5G*) and stimulates food intake (*Figure 5O*) in lean mice at baseline. Following six weeks of HFD exposure, we observed a clear reduction in this ability of ghrelin to activate AgRP neurons (*Figure 5H–K*) and drive feeding (*Figure 5P*). Interestingly, we found that a higher, supraphysiologic dose of ghrelin (0.5 mg/kg; *Figure 5—figure supplement 1E–H*) was able to induce normal calcium responses in AgRP neurons of DIO mice, indicating that the ghrelin resistance in AgRP neurons is partial. In contrast, ghrelin at this and even higher doses (1 mg/kg, *Figure 5P*) failed to have any effect on feeding in DIO animals. This indicates that the impairment of ghrelin-induced feeding in DIO mice is not solely due to reduced sensitivity of AgRP neurons, and may instead involve desensitization of downstream circuits, as suggested by our optogenetic manipulations (*Figure 3*). Subsequent weight loss did not restore the responsiveness of AgRP neurons in DIO mice to ghrelin, similar to observations with CCK (*Figure 5H–L*). Thus, obesity causes persistent resistance to two key gastrointestinal signals at the level of AgRP neuron activity, and this is associated with decreased ability of these hormones to modulate food intake.

## Discussion

Obesity arises due to an imbalance between energy intake and energy expenditure. Neural circuits in the hypothalamus are well known for monitoring energy balance, but how obesity alters the in vivo dynamics of these circuits has been unclear (*Andermann and Lowell, 2017*; *Rossi et al., 2019*). Here, we have used fiber photometry and optogenetics in conjunction with a mouse model of diet-

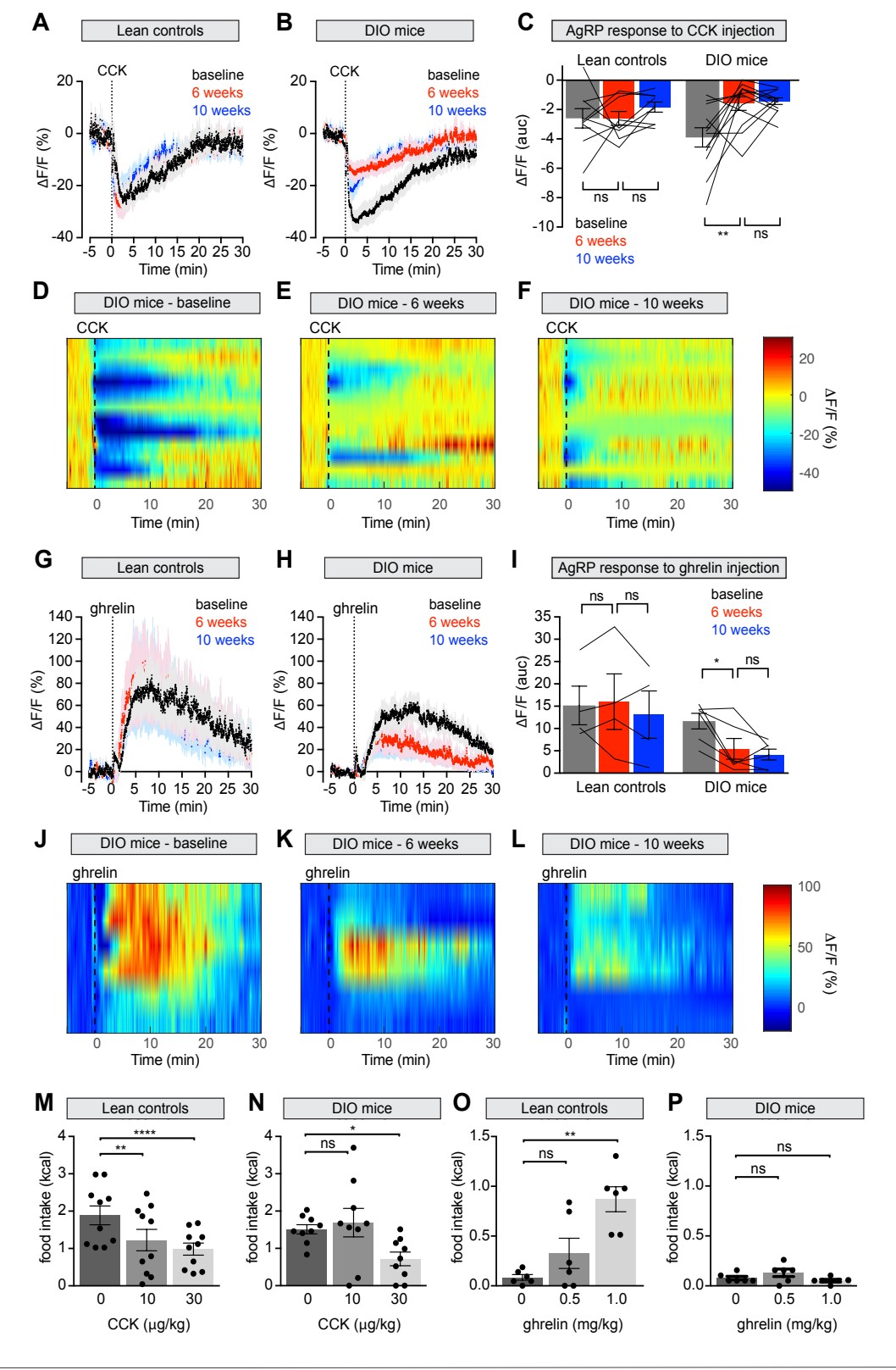

**Figure 5.** Diet-induced obesity attenuates AgRP neural and behavioral responses to CCK and ghrelin. (**A and B**) Calcium signal from AgRP neurons in fasted control (**A**) and DIO (**B**) mice treated with CCK 30 µg/kg IP at baseline (black), after 6 weeks on chow or HFD (red), and after an additional 4 weeks of chow (blue). (n = 9–12 mice per group). (**C**) Quantification of ΔF/F from (**A**) and (**B**) showing area under the curve (AUC) for 20 min after injection. (**D–F**) Peri-injection heatmaps depicting ΔF/F during photometry recording in individual DIO mice from (**B**) at baseline (**D**), after 6 weeks on

*Figure 5 continued*

HFD (**E**), and after 4 weeks recovery (**F**). (**G and H**) Calcium signal from AgRP neurons in ad libitum fed control (**G**) and DIO (**H**) mice treated with ghrelin 0.1 mg/kg IP at baseline (black), after 6 weeks on chow or HFD (red), and after an additional 4 weeks of chow (blue). (n = 4–6 mice per group). (**I**) Quantification of $\Delta F/F$ from (**G**) and (**H**) showing area under the curve (AUC) for 30 min after injection. (**J–L**) Peri-injection heatmaps depicting $\Delta F/F$ during photometry recording in individual DIO mice from (**H**) at baseline (**J**), after 6 weeks on HFD (**K**), and after 4 weeks recovery (**L**). (**M and N**) 30 min food intake in fasted control (**M**) or DIO (**N**) animals re-fed with chow or HFD, respectively following injection of the indicated doses of CCK. (n = 9–10 mice per group). (**O and P**) 30 min food intake in ad libitum fed control (**O**) or DIO (**P**) animals of chow or HFD, respectively, following injection of the indicated doses of ghrelin (n = 6 mice per group). *p<0.05, **p<0.01, and ****p<0.0001 as indicated. There was no significant difference between lean control and DIO groups at baseline. (**A,B,G,H**) Traces represent mean ± SEM. (**C,I**) Lines denote individual mice. Error bars represent mean ± SEM. (**M, N,O,P**) • denotes individual mice. Error bars represent mean ± SEM.

The online version of this article includes the following figure supplement(s) for figure 5:

**Figure supplement 1.** The effect of DIO on AgRP neuronal responses to CCK and ghrelin is dose-dependent.

induced obesity to study how feeding circuits are altered by chronic changes in diet. We have shown that obesity broadly desensitizes AgRP neurons to an array of nutritionally relevant stimuli and further alters the ability of AgRP neuron stimulation to drive feeding. Importantly, some of these changes persist after animals lose weight, revealing that dietary history can have long-lasting effects on feeding circuit dynamics. This metabolic hysteresis may contribute to difficulty of maintaining weight loss.

## Obesity selectively desensitizes feeding circuits to dietary fat

Meal size is regulated by the signals from the gut that tell the brain about the quantity and quality of recently ingested food (*Clemmensen et al., 2017*; *Cummings and Overduin, 2007*; *Williams and Elmquist, 2012*). While many studies have investigated how the levels of circulating hormones are altered by diet or obesity (*Lean and Malkova, 2016*), it remains unknown how these and other peripheral changes are reflected in the dynamics of feeding circuits in the brain.

To investigate this, we monitored the activity of AgRP neurons while infusing different macronutrients into the stomach of awake mice. We then longitudinally tracked these nutrient responses as animals gained weight due to a HFD and then lost weight following return to a low-fat diet. This revealed that diet-induced obesity blunts the inhibition of AgRP neurons by intragastric fat but has no effect on the response to intragastric sugar or protein (*Figure 4*, *Figure 4—figure supplement 1*). Ingestion of fat is communicated to AgRP neurons in part through CCK (*Beutler et al., 2017*) and, consistent with this, we found that obesity also desensitizes AgRP neurons to peripheral CCK (*Figure 5A–F*). Remarkably, this resistance to dietary fat and CCK persisted after mice had been returned to a low-fat diet and lost weight, which normalized their feeding behavior (*Figure 2B and D*) and neural responses to food cues (*Figure 1F and J–L*). This identifies a gastrointestinal fat→CCK→AgRP neuron pathway that is selectively and persistently impaired by HFD. Impairment of this pathway may contribute to the ability of dietary fat to drive obesity in mice and raises the possibility that diets with different macronutrient compositions may modulate homeostatic circuits in distinct ways.

## High-fat diet alters hunger circuit dynamics in ways predicted to both promote and suppress obesity

A fundamental question is whether diet-induced obesity is caused by the dysregulation of homeostatic circuits, or alternatively whether it occurs despite the action of those circuits to resist weight gain (*Ottaway et al., 2015*; *Timper and Brüning, 2017*; *Velloso and Schwartz, 2011*). By monitoring AgRP neurons in vivo during weight gain and loss, the experiments described here provide new insight into this question.

Overall, we find that HFD triggers changes in AgRP neuron regulation that would be predicted to both promote and suppress obesity. Among the former is the finding that obesity renders AgRP neurons desensitized to intragastric fat and CCK. This change would create a positive feedback loop whereby over-ingestion of fat would reduce the ability of subsequent fat intake to attenuate feeding, thereby promoting weight gain. How exactly information about gastrointestinal fat is relayed to AgRP neurons, and where in the pathway from the gut to the hypothalamus resistance to CCK develops, remain unknown. However, plausible candidate mechanisms include desensitization of CCK-

responsive vagal afferents (*Daly et al., 2011*; *Grabauskas et al., 2019*) and brain stem neurons (*Covasa et al., 2000*; *Wall et al., 2019*) as well as changes in lipid-induced CCK release (*Kentish and Page, 2015*).

We also found that obesity blunts the rapid inhibition of AgRP neurons that occurs when hungry mice see and smell food (*Figure 1*). While the function of this rapid sensory inhibition of AgRP neurons remains unresolved (*Chen and Knight, 2016*), its magnitude correlates with the amount of food subsequently consumed; that is, the more AgRP neurons are inhibited when food is presented, the more food the mouse consumes in the ensuing meal (*Beutler et al., 2017*). Consistent with this, we found that the diminished inhibition of AgRP neurons by food presentation in obese mice was associated with less subsequent food consumption (*Figure 2B and D*). After return to a low-fat diet and weight loss, the sensory response to chow was restored (*Figure 1F*) as was chow intake (*Figure 2B*). This indicates that the prediction of food intake that is relayed to AgRP neurons reliably anticipates how weight gain and loss will alter feeding behavior.

We also identified changes in AgRP circuit dynamics that would be predicted to suppress weight gain. Most notably we found that obesity blunted the activation of AgRP neurons by the appetite promoting hormone ghrelin (*Figure 5G–L*). This finding is consistent with prior work showing that AgRP neurons from obese mice are resistant to activation by ghrelin in slice and that obese mice show reduced Fos in the arcuate nucleus and eat less food following peripheral ghrelin injection (*Briggs et al., 2010*; *Briggs et al., 2014*; *Perreault et al., 2004*). Interestingly, our finding that AgRP neurons become resistant to both ghrelin and CCK, along with substantial prior evidence of leptin resistance in obesity (*Myers et al., 2010*), suggests that AgRP neurons may become generally resistant to hormonal input following weight gain. It will be important to investigate whether there is a common mechanism by which AgRP neurons become desensitized to these diverse signals, and whether this change is adaptive or deleterious.

## Diet-induced obesity alters the behavioral response to AgRP neuron stimulation

To understand the functional significance of these changes in AgRP neuron dynamics, we also measured the behavioral response to optogenetic manipulation of AgRP neurons before and after the development of obesity. The most prominent effect we observed was that optogenetic stimulation was able to partially rescue the deficit in food consumption of obese mice following an overnight fast (*Figure 3D,E*). This indicates that downstream circuits retain the ability to respond to AgRP neuron stimulation in obese mice. It also suggests that the reason why obese mice show diminished fasting-induced re-feeding may be that AgRP neurons are not fully activated by food deprivation in these animals (and that optogenetic stimulation supplies this missing activation). This interpretation is consistent with prior data showing DIO mice have reduced arcuate Fos in response to fasting (*Briggs et al., 2011*) and an inability to respond to ghrelin (*Figure 5G–L and O,P*, and *Briggs et al., 2010*). It is also consistent with the tendency of elevated leptin levels in obese mice to suppress the activation of AgRP neurons by fasting (*Becskei et al., 2010*).

While these observations point to a primary defect in AgRP neuron activation, our findings also suggest that downstream circuits may have reduced sensitivity to AgRP neuron stimulation in obese animals. For example, we found that concurrent stimulation had a stronger effect on food intake than pre-stimulation in DIO mice (*Figure 3B,C,E*) whereas there was no difference between these two protocols in lean animals (*Figure 3B–E*). Given that concurrent stimulation causes stronger activation of this circuitry (*Burnett et al., 2019*), this suggests that greater AgRP neuron activity is required to drive food intake in obese mice. This observation is also consistent with the possibility that NPY signaling, which is necessary for food intake in response to pre- but not concurrent stimulation, is attenuated in DIO animals (*Beck, 2006*; *Chen et al., 2019*; *Stricker-Krongrad et al., 1994*; *Stricker-Krongrad et al., 1997*).

One limitation of these findings is that fiber photometry cannot measure absolute firing rates. Thus, we have described how obesity changes the response of AgRP neurons to food cues, nutrients infusions, and hormone injections, but we cannot measure how the tonic activity of AgRP neurons changes over time. Slice electrophysiology studies have suggested that the activity of AgRP neurons may be elevated in obese animals and then resistant to further increases caused by fasting (*Baver et al., 2014*; *Wei et al., 2015*). This could be tested by using extracellular recordings to record spiking activity in vivo (*Mandelblat-Cerf et al., 2015*). Another important question not

addressed here is whether there is heterogeneity in AgRP neuron responses to diet-induced obesity. Such heterogeneity could be investigated using single-cell calcium imaging or electrophysiology.

## Dietary history has persistent effects on feeding circuit dynamics

One reason why obesity is such a challenging clinical problem is that patients experience great difficulty maintaining weight loss (*Fothergill et al., 2016*; *Leibel et al., 2015*). To explain this, it has been suggested that obesity may alter the body weight set point, such that the energy homeostasis system comes to defend a new, heavier body weight (*Leibel et al., 1995*). In mice, diet-induced obesity can be reversed by restricting animals' caloric intake, but these formerly obese animals remain sensitized to accelerated weight gain when returned to ad libitum chow or HFD (*Briggs et al., 2013*; *Schmitz et al., 2016*). This recidivism has been associated with a variety of physiologic mechanisms (*Friedman, 2002*; *Ravussin et al., 2018*; *Zigman et al., 2016*), but a core idea is that homeostatic circuits are altered by obesity in a way that persists after weight loss and promotes weight re-gain.

In this study, we set out to look for evidence of such hysteresis in the dynamics of AgRP neurons. We found that that exposure to HFD can impair to neural responses to nutrients and hormones in a way that would be predicted to promote weight gain, and that this impairment persists for weeks after mice have been returned to a low-fat diet and lost weight. This reveals that dietary history can have long-lasting effects on the dynamics of the neural circuits that control hunger (*Matikainen-Ankney and Kravitz, 2018*). An important task for the future will be to clarify the molecular basis of this diet-induced resistance of AgRP neurons to gastrointestinal hormones and nutrients, since this may suggest novel strategies for treating obesity.

# Materials and methods

**Key resources table**

| Reagent type (species) or resource | Designation | Source or reference | Identifiers | Additional information |
|---|---|---|---|---|
| Strain, strain background (*Mus musculus*) | Wildtype | Jackson Labs | Stock No:000664 | |
| Strain, strain background (*M. musculus*) | AgRP-Cre | Jackson Labs | Stock No:012899 | |
| Strain, strain background (*M. musculus*) | ROSA26-lox Stoplox-ChR2-eYFP | Jackson Labs | Stock No:012569 | |
| Strain, strain background (adeno-associated virus-1) | AAV1.CAG. Flex.GCaMP6s | Addgene | ID:100842 | |
| Peptide, recombinant protein | CCK | Bachem | Product No: 4033010.0001 | |
| Peptide, recombinant protein | Ghrelin | R and D Systems | Catalog #:1465/1 | |
| Commercial assay or kit | Leptin ELISA | Crystal Chem | Catalog #:90030 | |
| Commercial assay or kit | Insulin ELISA | Crystal Chem | Catalog #:90080 | |
| Chemical compound, drug | 'high-fat diet; HFD' | Research Diets | Diet Formula: D12492 | |
| Chemical compound, drug | 'low-fat pellets; food pellets' | Bio-Serv | Product #:F0163 | |
| Software, algorithm | MATLAB | MathWorks | RRID:SCR_001622 | |
| Software, algorithm | Prism | GraphPad | RRID:SCR_002798 | |

## Animals

Experimental protocols were approved by the University of California, San Francisco IACUC following the National Institutes of Health guidelines for the Care and Use of Laboratory Animals (Protocol# AN179674). Animals were housed in 12 hr dark/light cycle with ad libitum access to food and water. Animals were maintained on ad libitum chow (LabDiet 5053, Fort Worth, TX) or high-fat diet (HFD; Research Diets D12492, New Brunswick, NJ), and were fasted for 16 hr before experiments as indicated in the text and figures. They maintained ad libitum access to water throughout this time. *Agrp^tm1(cre)Lowl*(AgRP-Cre, #012899, Jackson Labs, Bar Harbor, ME) animals used in all fiber photometry experiments have been previously described and have been backcrossed onto a C57BL/6 background. To achieve channelrhodopsin-2 expression in AGRP neurons for optogenetic experiments, Agrp-Cre mice were crossed with 129S-*Gt(ROSA)26Sor^tm32(CAG-COP4*H134R/EYFP)Hze* (ROSA26-loxStoplox-ChR2-eYFP, #012569, Jackson Labs) to generate double mutant animals (AgRP::ChR2). For experiments examining the effects of CCK and ghrelin on acute feeding behavior, and for studies examining the effects of plasma leptin and insulin levels on fast re-feeding in DIO animals, C57BL/6J (wildtype #000664, Jackson Labs) mice were used. No statistical methods were used to determine sample sizes. Male and female mice ranging from 8 to 20 weeks were used. Animals used in intragastric infusion experiments were individually housed, as were animals used in hormone-induced feeding experiments. All other mice were group-housed.

## Diet-induced obesity

All baseline fiber photometry recordings and optogenetics experiments were performed on animals fed a chow diet. Following baseline fiber photometry or optogenetic experiments, DIO animals were placed on HFD. For photometry experiments, animals were assigned to DIO or lean control groups to match baseline body weight between groups. Photometry and fast re-feeding experiments were performed in DIO and lean control animals after 4–9 weeks on their respective diets as indicated in the text and figures, after which DIO animals were returned to a standard chow diet for 4 weeks and neuronal recordings and fast re-feeding experiments were repeated. DIO and control animals underwent the same fiber photometry experiments at the same time points to control for fluctuations in calcium-induced fluorescence over time. Optogenetics experiments were performed at baseline and after 3–5 weeks on HFD in the same cohort of animals. Hormone-induced feeding experiments and plasma hormone studies were performed at a single time point on separate DIO and control animals after 4–8 weeks on HFD.

## Stereotaxic surgery

For photometry experiments, we used recombinant AAV expressing cre-dependent GCaMP6s (AAV1.CAG.Flex.GCaMP6s, Penn Vector Core/Addgene, Philadelphia, PA/Watertown, MA). AAV was stereotaxically injected unilaterally above the arcuate nucleus (ARC) of AgRP-Cre, mice. During the same surgery a commercially available photometry cannula (MFC_400/430–0.48_6.1 mm_MF2.5_FLT, Doric Lenses, Franquet, Quebec) was implanted unilaterally in the ARC at the coordinates x = −0.3 mm, y = −1.85 mm, z = −5.9 mm from bregma. Mice were allowed 2–4 weeks for viral expression and recovery from surgery before photometry recording or intragastric catheter implantation.

For optogenetic experiments, commercially available fiberoptic implants (MFC_200/245_0.37_6.1 mm_ZF1.25_FLT, Doric Lenses) were placed unilaterally above the arcuate nucleus of AgRP::ChR2 mice at the coordinates x = −0.30 from bregma, y = −1.8 from bregma, z = −5.7 from dorsal skull surface. Mice were allowed 1 week recovery from surgery before behavior experiments.

## Intragastric catheter implantation

Intragastric catheters were made and implanted as described in detail previously (*Beutler et al., 2017*; *Ueno et al., 2012*). Catheters were constructed by attaching 8 cm of Silastic tubing (508–003, Silastic) and 8 cm Tygon tubing (Tygon, AAD04119) to opposite ends of a curved metal connector (NE-9019, Component Supply Company, Sparta, TN). A 1 cm circle of biologically compatible mesh (gifted by Raul Lazaro) was attached to the silastic tubing 2.3–2.8 cm distal to the edge of the metal connector using adhesive (Xiameter RTV-3110 base and Dow Corning four catalyst). A 1 cm by 1.5 cm oval of felt was affixed to the silastic tubing at the distal edge of the curve in the connector and

a 0.5 cm by 1 cm strip of felt was affixed around the metal connector on the proximal edge of its curve. A luer adaptor was placed into the free end of the Tygon tubing (LS20, Instech, Plymouth Meeting, PA). Assembled catheters were sterilized using ethylene oxide.

AgRP-Cre mice with functional photometry implants were anesthetized with ketamine/xylazine and the surgical areas shaved and scrubbed with betadine and alcohol. A skin incision of about 1 cm was made between the scapula and the skin dissected from the subcutaneous tissue toward the left flank. A midline abdominal skin incision about 1.5 cm was made extending from the xyphoid process caudally and the skin was dissected from the subcutaneous tissue toward the left flank to complete a subcutaneous tunnel between the two incisions. A hemostat was used to pull the sterilized catheter through the tunnel. The linea alba was incised and the abdominal cavity entered. A small incision was made in the left lateral abdominal wall through which the intragastric catheter was passed into the abdominal cavity. The stomach was externalized and a small puncture made using a jeweler's forceps. The tip of the catheter was immediately placed into the puncture site and sutured into place using the felt circle with polypropylene suture. Saline injection into catheter confirmed absence of leakage. The stomach was placed back in the abdominal cavity, which was washed with sterile saline. The abdominal muscle was sutured and the skin incision closed in two layers. Next, the catheter was secured at its interscapular site with sutures into the felt oval and surrounding muscle. Finally, the interscapular skin incision was closed. Post-operatively, mice were treated with enrofloxacin, normal saline, and buprenorpine and allowed 10–14 days to recover prior to intragastric infusion and photometry experiments.

## Fiber photometry

Two rigs for performing fiber photometry recordings were constructed following basic specifications previously described with minor modifications (*Chen et al., 2015*; *Gunaydin et al., 2014*). A 473 nm laser diode (Omicron Luxx) was used as the excitation source. This was placed upstream of an optical chopper (MC2000, Thorlabs, Newton, NJ) that was run at 400 Hz. The laser was then split with a beam splitter (CM1BS013, Thorlabs) and the slit laser beams each reflected off two kinetic mirrors (BB1-E02, KM100, Thorlabs) to allow adjustment of the light path. Each laser beam was passed through a GFP excitation filter (MF469-35, Thorlabs), reflected by a dichroic mirror (FF495-Di03−25 × 36, Semrock, Rochester, NY) and coupled through a fiber collimation package (F240FC-A, Thorlabs) into a commercially available patchcord (MFP_400/460/1100–0.48_2 m_FCM-MF2.5 or MFP_400/430/1100–0.48_2 m_FCM-MF2.5, Doric Lenses). Patchcords were not changed within experiments. The patchcords were then linked to fiberoptic implants through bronze sleeves (SLEEVE_BR_2.5, Doric Lenses). Fluorescence outputs were each filtered through a GFP emission filter (MF525-39, Thorlabs) and focused by convex lenses (LA1255A, Thorlabs) onto photoreceivers (2151, Newport, Irvine, CA). The signals were output into lock-in amplifiers (SR810, Stanford Research System, Sunnyvale, CA) with time constant at 30 ms to allow filtering of noise at higher frequency. Those two lock-in amplifiers receive the frequency signal of chopper split by BNC splitter. Signals were then digitized with a LabJack U6-Pro and recorded using software provided by LabJack (https://labjack.com/support/software) with 250 Hz sampling rate.

All photometry experiments were performed in operant chambers (H10-11M-TC, Coulbourn, Holliston, MA) inside a sound-attenuating cubicle (ENV-022MD, Med Associates, Fairfax, VT). Experiments were performed during the dark cycle in a dark environment. Mice that did not show a sensory response to chow of at least ΔF/F 20% at baseline were assumed to be technical failures and were excluded from further experiments or analysis.

To minimize contamination of the signal by dust in the light path, we cleaned the fiberoptic on the mouse with connector cleaning sticks (MCC25, Thorlabs) and 70% ethanol before each recording. A syringe needle was used to pick out debris that occasionally became stuck in the sleeve. We also refrained from re-aligning light path of photometry rig to ensure consistency.

## Intragastric infusions

Nutrients were infused via intragastric catheters using a syringe pump (70–2001, Harvard Apparatus, Holliston, MA) similar to what has been described previously (*Beutler et al., 2017*) All infusions were delivered at 100 µL per min with a total infusion volume of 1.2 mL and infusion time of 12 min. All photometry experiments involving intragastric infusion were performed in fasted animals. Animals

were habituated to behavioral chambers for 20 min during photometry recording. During this time, the intragastric catheter was attached to the syringe pump using plastic tubing and adapters (AAD04119, Tygon; LS20, Instech). Photometry recording was continued for 15 min after the end of infusion. One to three trials of the same experiment for each mouse were combined, averaged, and treated as a single replicate. For peristimulus plots, time zero was defined as the moment that the infusion pump started.

All nutrients were diluted into deionized water fresh for each experiment. Intralipid (I141, Sigma-Aldrich, St. Louis, MO) was used undiluted; unflavored premium collagen peptides (Sports Research) and glucose were dissolved at 0.24 g/mL.

## Hormone injections

Hormones were injected at the concentrations and routes indicated below during photometry recording and prior to measurement of food intake. All compounds were injected at a volume of 10 µL/g body weight. For photometry experiments, animals were habituated to the recording chambers for 20 min prior to injection. Following hormone injection, photometry recording continued for 35 min. One to four trials of the same experiment for each mouse were combined, averaged, and treated as a single replicate. For peristimulus plots, time zero was defined as the moment that the investigator opened the behavioral chamber. For feeding experiments, animals in their home cage were injected with hormone immediately before the measurement of food intake began.

We used the following doses: CCK octapeptide 30 µg/kg or 10 µg/kg IP (4033010, Bachem, Torrance, CA) and ghrelin 1.0 mg/kg, 0.5 mg/kg, or 0.1 mg/kg IP (1465/1, R and D Systems, Minneapolis, MN) as indicated in the text and figures.

## Food presentation

To minimize the effects of novelty and to ensure that mice knew presented objects were palatable food items, all mice including chow fed control animals were exposed prior to photometry testing or measurement of food intake during fast re-feeding to HFD or chocolate for 1–2 nights. For fiber photometry experiments, mice were fasted overnight (16 hr), acclimated to the behavioral chamber, and then presented with chow or HFD during fiber photometry recording. For peristimulus plots time zero was defined as the moment that the investigator opened the behavioral chamber. For fast re-feeding experiments, mice were fasted overnight (16 hr) then presented with chow, HFD, or chocolate and food intake monitored after 30 min. For hormone-induced feeding experiments, mice were fasted (CCK) or ad libitum fed (ghrelin) and food intake was measured 30 min (CCK) or 120 min (ghrelin) following injection based on previous studies (*Briggs et al., 2010*; *Essner et al., 2017*). One to three trials of the same experiment for each mouse were combined, averaged, and treated as a single replicate.

## Optogenetic feeding behavior

Optogenetic stimulation was performed as previously described (*Chen et al., 2016*; *Chen et al., 2019*). A 473 nm laser was passed through a TTL signal generator (H03-14, Coulbourn) and synchronized with the pellet distribution system (H14-01M-SP04 and H14-23M, Coulbourn) through the Coulbourn Graphic State software. The laser was passed through a single patch cable (Doric Lenses) to a custom fiber optic patch cable (FT200UMT, CFLC230-10, Thorlabs; F12774, Fiber Instrument Sales) through a rotary joint (FRJ 1 × 1, Doric Lenses). Patch cables were attached to the mouse implants by a zirconia mating sleeve (ADAL1, Thorlabs). Laser power was set to 15–20 mW at the terminal of each patch cable. The laser was modulated at 20 Hz on a 2 s ON and 3 s OFF cycle with a 10 ms pulse width.

Following implant surgery, mice were given seven days to recover before experiments. During this recovery they were given ad libitum chow and supplied with ad libitum food pellets (20 mg F0163, Bio-Serv, Flemington, NJ) or 3 g of high-fat diet (D12492, Research Diets) for 1–3 days before testing. Mice were habituated to the behavior chamber (H10-11M-TC with H10-11M-TC-NSF, Coulbourn) and pellet distribution system for 15 hr before the first experiment. Experiments were conducted during the light cycle.

All laser stimulation protocols follow this general structure: 30 min habituation followed by 30 min of access to pellets or high-fat diet. For the no-stimulation protocol mice were given 30 min of

habituation followed by 30 min of food access without laser stimulation. For the concurrentstimulation protocol, mice were given 30 min of habituation followed by 30 min of food access with laser stimulation. For the pre-stimulation protocol, mice were given 30 min of habituation with laser stimulation and no food access, then 30 min of food access without laser stimulation (*Chen et al., 2019*).

Each of the above stimulation protocols was performed in animals in the fasted and fed state, and with food access to pellets or HFD on separate days. For experiments with food pellets, the pellet distribution system detected pellet removal from the food hopper using a built-in photosensor (H20-93, Coulbourn) and a pellet was dispensed 10 s after each removal. For experiments with HFD access, a single 3 g piece of HFD was placed in the behavior chamber at the end of habituation and weighed before and after 30 min of food access.

All experiments were performed at baseline and after development of DIO with 3–5 weeks of ad libitum access to HFD.

### CCK and Ghrelin-induced feeding experiments

Wildtype mice were individually housed with access to ad libitum chow or HFD in their home-cage for 8 weeks and habituated to IP injection of saline for 1 day before experiments. To monitor CCK-induced feeding suppression, DIO and control mice were fasted for 16 hr and at the onset of dark cycle injected with vehicle, CCK 10 µg/kg or CCK 30 µg/kg on separate nights then immediately presented with chow (control animals) or HFD (DIO animals). Food intake was monitored at 30 min after injection. To monitor ghrelin-induced feeding stimulation, ad libitum fed DIO and control mice were injected with vehicle, ghrelin 0.5 mg/kg, or ghrelin 1.0 mg/kg at the onset of light cycle on separate days. Chow (control animals) or HFD (DIO animals) intake was monitored at 120 min after injection. Order of doses was varied among the cohorts.

### Plasma hormone studies

Wildtype mice were given access to ad libitum chow or HFD in their home-cage for 4 weeks, then fasted overnight before blood collection via submandibular venipuncture into EDTA-coated tubes for plasma separation. ELISA assays for plasma leptin (90030, Crystal Chem) and insulin (90080, Crystal Chem) were performed according to the manufacturer's instructions. For the leptin assay, all samples were diluted 1:5 to ensure all measurements remained within the standard range.

### Quantification and statistical analysis

#### Photometry analysis

Data were analyzed using a custom MATLAB script as described previously (*Beutler et al., 2017*; *Chen et al., 2015*). For intragastric infusion and IP hormone injection experiments, background fluorescence was corrected by subtracting the photometry signal in the absence of mice from total signal. Data were then low-pass filtered at 0.5 Hz due to their slow and sustained change in response to stimuli and down sampled to 10 Hz. For peri-stimulus time plots the median value of data points in a 2 min window flanking the $-5$ min time point before each treatment was used as the normalization factor (F0) to calculate $\Delta F(t)/F0 = (F(t)-F0)/F0$. To calculate the change of fluorescent signal at indicated time points after treatment, all data points F(t) over the indicated time range were averaged as Fa to estimate $\Delta Fa/F0 = (Fa-F0)/F0$.

For all experiments correction for photobleaching was not necessary due to the low laser power used during photometry recordings (0.07 mW), the short time windows for experiments (around 60 min), and the fact that all experimental groups had control groups treated with identical laser powers. In addition, correction for photobleaching could easily cause over-correction due to the slow and sustained effect in our experiments.

#### Behavior data analysis

Optogenetics experiments in which mice were fed pellets were analyzed using a custom MATLAB script. Consumption of each pellet was defined as the first pellet removal event after each food pellet delivery. Total food consumption was estimated by subtracting the pellets found dropped after each experiment from the total number of food removal events.

## Statistical analysis

Fiber photometry data were subjected to analysis as described above. For traces in *Figures 1*, *4* and *5*, and associated figure supplements, ΔF/F(%) represents the mean $\Delta F(t)/F0^*100$. Bar graphs depicting the sensory response to chow and the response to nutrient infusion in *Figures 1* and *4*, and associated figure supplements show the mean ΔF/F(%) over a 5 min time window following food presentation (*Figure 1*) or at the end of intragastric infusion (*Figure 4* and supplement). Bar graphs depicting the response of AgRP neurons to hormone injection in *Figure 5* and associated supplement show the area under the curve for the time intervals indicated in the figure legends.

The effects of DIO and subsequent return to chow diet on body weight and fluorescence changes in response to the sensory detection of food, hormones, and nutrients were analyzed using two-way, repeated-measures ANOVA, as were the effects of DIO on optogenetic stimulation of feeding. The effects of fasting, CCK and ghrelin on feeding in lean control and DIO animals were analyzed using one-way, repeated-measures ANOVA. The effect of DIO on chocolate consumption, fast re-feeding and the effects of DIO on plasma leptin and insulin levels in *Figure 2—figure supplement 1* were analyzed using an unpaired T-test. The Holm-Sidak multiple comparisons test was used in conjunction with ANOVA. Linear regression analysis was used to examine correlation between weight change and changes in neuronal activity or between food intake during fast re-feeding and plasma hormone level (*Figure 1—figure supplement 1*, *Figure 2—figure supplement 1*, and *Figure 5—figure supplement 1*). All statistical analysis was performed using Prism. Numbers of animals (biological replicates) used in each experiment are included in the figure legends. Where multiple trials of the same experiment were performed, these technical replicates were averaged and then treated as a single replicate.

Significance was defined as $p<0.05$ and is indicated on figures and in figure legends.

## Acknowledgements

LRB is supported by NIH grants K08DK118188 and P30 DK063720. YC is supported by a Howard Hughes Medical Institute International Student Fellowship. ZAK is a Howard Hughes Medical Institute Investigator and this work was supported by the New York Stem Cell Foundation, American Diabetes Association Pathway Program, Rita Allen Foundation, McKnight Foundation, Alfred P Sloan Foundation, Brain and Behavior Research Foundation, Esther A and Joseph Klingenstein Foundation, UCSF Program for Breakthrough Biomedical Research, UCSF Diabetes Center, UCSF Nutrition Obesity Research Center, and NIH grants DP2-DK109533, R01DK106399, and R01NS094781 (to ZAK).

## Additional information

### Funding

| Funder | Grant reference number | Author |
| --- | --- | --- |
| National Institutes of Health | R01DK106399 | Zachary A Knight |
| National Institutes of Health | R01NS094781 | Zachary A Knight |
| National Institutes of Health | DP2DK021153 | Zachary A Knight |
| Howard Hughes Medical Institute | Investigator | Zachary A Knight |
| American Diabetes Association | Pathway Award | Zachary A Knight |
| New York Stem Cell Foundation | Robertson Investigator Award | Zachary A Knight |
| Rita Allen Foundation | Scholar Award | Zachary A Knight |
| National Institutes of Health | K08DK118188 | Lisa R Beutler |
| National Institutes of Health | P30 DK063720 | Lisa R Beutler |
| McKnight Foundation | | Zachary A Knight |
| Alfred P. Sloan Foundation | | Zachary A Knight |

| Brain and Behavior Research Foundation | Zachary A Knight |
| Esther A. and Joseph Klingenstein Fund | Zachary A Knight |
| Diabetes Center at UCSF | Lisa R Beutler |
| UCSF Nutrition and Obesity Research Center | Zachary A Knight |

The funders played no role in the design or interpretation of the work.

## Author contributions
Lisa R Beutler, Conceptualization, Data curation, Formal analysis, Supervision, Investigation, Methodology, Writing - original draft, Writing - review and editing; Timothy V Corpuz, Data curation, Investigation, Writing - original draft; Jamie S Ahn, Seher Kosar, Weimin Song, Investigation; Yiming Chen, Data curation, Formal analysis; Zachary A Knight, Conceptualization, Resources, Supervision, Funding acquisition, Writing - original draft, Project administration, Writing - review and editing

## Author ORCIDs
Lisa R Beutler (iD) https://orcid.org/0000-0002-9489-8098
Zachary A Knight (iD) https://orcid.org/0000-0001-7621-1478

## Ethics
Animal experimentation: Experimental protocols were approved by the University of California, San Francisco IACUC following the National Institutes of Health guidelines for the Care and Use of Laboratory Animals. (protocol# AN179674).

## Decision letter and Author response
Decision letter https://doi.org/10.7554/eLife.55909.sa1
Author response https://doi.org/10.7554/eLife.55909.sa2

## Additional files

### Supplementary files
• Transparent reporting form

### Data availability
All data generated or analysed during this study are included in the manuscript and supporting file.

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
