## [Decision Letter]

**Acceptance summary:**

The inclusion of additional experiments and your point by point responses have convinced us that your paper provides novel data that will be of interest to the readers of *eLife*.

**Decision letter after peer review:**

Thank you for submitting your article "Obesity causes selective and long-lasting desensitization of AgRP neurons to dietary fat" for consideration by *eLife*. Your article has been reviewed by three peer reviewers, one of whom is a member of our Board of Reviewing Editors, and the evaluation has been overseen by Catherine Dulac as the Senior Editor. The reviewers have opted to remain anonymous.

The reviewers have discussed the reviews with one another and the Reviewing Editor has drafted this decision to help you prepare a revised submission.

As the editors have judged that your manuscript is of interest, but as described below that additional experiments are required before it is published, we would like to draw your attention to changes in our revision policy that we have made in response to COVID-19 (https://elifesciences.org/articles/57162). First, because many researchers have temporarily lost access to the labs, we will give authors as much time as they need to submit revised manuscripts. We are also offering, if you choose, to post the manuscript to bioRxiv (if it is not already there) along with this decision letter and a formal designation that the manuscript is 'in revision at *eLife*'. Please let us know if you would like to pursue this option. (If your work is more suitable for medRxiv, you will need to post the preprint yourself, as the mechanisms for us to do so are still in development.)

Summary:

Beutler and colleagues describe results from a series of studies investigating the effects of diet-induced obesity on the responses of AgRP neurons to a number of physiological stimuli including intragastric infusions and the hormones ghrelin and CCK. The manuscript describes real-time activity of AgRP neurons in response to exterosensory cues, intragastric-infused nutrients and peripheral-injected hormones, suggesting that diet-induced obesity blunts AgRP neuron response to food cues, selectively to intragastric fat and injection of CCK and ghrelin. Further, activation on AgRP neurons using two different stimulation protocols and found that fast-induced activation is attenuated by obesity, which was rescued by the opto-stimulation. Overall the studies are well described, provide novel information and are potentially of wide interest.

Essential revisions:

A question arises regarding data demonstrating that DIO exposure inhibits fasting induced hyperphagia, and that the anticipatory dynamics of AgRP neurons track these behavioral fluctuations. While the anticipatory dynamics appear to mostly (not in Figure 1I) correlate with changes in feeding, it's unclear if this activity change is required for the differing feeding patterns. In the current study, the authors demonstrate that DIO blunts the suppression of ΔF/F in AgRP neurons of mice presented with either chow or HFD (Figure 1). This implies the neurons are less inhibited in response to anticipatory cues after DIO. This increased activity appears further supported by slice work showing that NPY neurons are activated in DIO (Baver et al., 2014, Wei et al., 2015). If AgRP neurons are more active in DIO and after food presentation (as observed by the blunting of the ΔF/F), then one may predict DIO mice to increase feeding. However, in Figure 2B and D the authors show that DIO mice have a blunted chow or HFD food intake after an overnight fast (which is similar to previous reports). Therefore, it's unclear if the anticipatory dynamic activity state of AgRP neurons is required for the feeding behavior. A dissociation of this activity with feeding is further supported with the data presented in Figure 1I and Figure 2D. Specifically, while the ΔF/F remains blunted to anticipatory cues in the DIO group after 4 weeks on chow (Figure 1I), however this group returns to the food intake levels observed at baseline (Figure 2D). Possibly an explanation lies in work showing that the fasting induced activation of NPY neurons is blunted in DIO mice (Baver et al). However these data are likely separate from anticipatory/sensory dynamics, which could question the role of the anticipatory dynamics in the feeding response. This topic could be better addressed/explained/detailed more in the manuscript. How do these dynamics link to feeding behavior? How are these dynamics similar/different to gastrointestinal/hormonal/nutrient cues?

Similarly, what is the baseline control for DIO? Are these mice fed HFD for any period of time or are they fed chow for baseline and then switched to HFD afterward? If the baseline group is on HFD, how long were they fed HFD and considered baseline? This is important given there is considerable variability in the baseline groups in several panels possibly contributing to the perceived changes (e.g. Figure 1G-I; Figure 3; Figure 4E-G; Figure 5A-C; Figure 5E-G; Figure 4—figure supplement 1A-D). What accounts for this variability? The significance or lack of significance for Figures 4G, 5C, and possibly others appear to have benefited from this variability. Similar to the previous point; statistical comparisons should be made within and across groups at different times to assess these changes.

Overall, the manuscript is somewhat incremental. An exception is the fact that some of the attenuated responses to nutrients/food cues are long lasting after reintroduction of LFD for few weeks. A pair-fed control group would be critical to investigate the extent to which is the diet, and not the obesity itself, that causes the phenotypes described. Quantification of fat mass (and not only body weight) and other metabolic parameters related to DIO are important. The manuscript falls short in their analysis of the different components of DIO that can be contributing to the observed phenotypes.

---

## [Author Response]

Essential revisions:A question arises regarding data demonstrating that DIO exposure inhibits fasting induced hyperphagia, and that the anticipatory dynamics of AgRP neurons track these behavioral fluctuations. While the anticipatory dynamics appear to mostly (not in Figure 1I) correlate with changes in feeding, it's unclear if this activity change is required for the differing feeding patterns. In the current study, the authors demonstrate that DIO blunts the suppression of ΔF/F in AgRP neurons of mice presented with either chow or HFD (Figure 1). This implies the neurons are less inhibited in response to anticipatory cues after DIO. This increased activity appears further supported by slice work showing that NPY neurons are activated in DIO (Baver et al., 2014, Wei et al., 2015). If AgRP neurons are more active in DIO and after food presentation (as observed by the blunting of the ΔF/F), then one may predict DIO mice to increase feeding. However, in Figure 2B and 2D the authors show that DIO mice have a blunted chow or HFD food intake after an overnight fast (which is similar to previous reports). Therefore, it's unclear if the anticipatory dynamic activity state of AgRP neurons is required for the feeding behavior.

Our data show that AgRP neurons of DIO mice are less inhibited by food presentation and therefore may be more active during feeding. The reviewer notes that this does not explain why DIO mice eat less food after fasting (they would be predicted to eat more). We agree. The sensory inhibition of AgRP neurons is correlated with subsequent food intake but does not cause that food intake.

It is important to emphasize that the paradoxical inhibition of AgRP neurons by sensory cues is observed in both lean and obese animals (Beutler et al., 2017; Chen et al., 2015) and its function is unresolved. We note this in the text (subsection “High-fat diet alters hunger circuit dynamics in ways predicted to both promote and suppress obesity”, third paragraph) and cite a review article we have written that discusses various hypotheses about the function of this response (Chen and Knight, 2016).

In the revised text, we have also included a citation of slice electrophysiology studies showing that AgRP neurons are more active in DIO mice as well as an extended discussion of some of the limitations of photometry, including the inability to determine absolute activity levels (subsection “Diet-induced obesity alters the behavioral response to AgRP neuron stimulation”, last paragraph).

A dissociation of this activity with feeding is further supported with the data presented in Figure 1I and Figure 2D. Specifically, while the ΔF/F remains blunted to anticipatory cues in the DIO group after 4 weeks on chow (Figure 1I), however this group returns to the food intake levels observed at baseline (Figure 2D). Possibly an explanation lies in work showing that the fasting induced activation of NPY neurons is blunted in DIO mice (Baver et al). However these data are likely separate from anticipatory/sensory dynamics, which could question the role of the anticipatory dynamics in the feeding response. This topic could be better addressed/explained/detailed more in the manuscript. How do these dynamics link to feeding behavior? How are these dynamics similar/different to gastrointestinal/hormonal/nutrient cues?

As stated above, we believe that the anticipatory dynamics of AgRP neurons are correlated with subsequent food intake but do not determine the amount of food consumed. The reviewer correctly notes that one exception to this correlation is Figure 1I, where the neural response of mice to HFD presentation is not restored after 4 weeks on chow (whereas food intake is restored). We have not speculated about possible causes for this discrepancy, in part because we are uncertain about whether it is a real difference (i.e. examination of the individual data points in Figure 1I reveals that 6/9 mice do show a rebound in responsiveness after return to chow, although the group means are not significantly different).

Similarly, what is the baseline control for DIO? Are these mice fed HFD for any period of time or are they fed chow for baseline and then switched to HFD afterward? If the baseline group is on HFD, how long were they fed HFD and considered baseline?

Mice in all experiments underwent the same protocol, which is depicted in Figure 1A, described in the first paragraph of the subsection “Diet-induced obesity attenuates the AgRP neuron response to the sensory detection of food”, and described in more detail in the Materials and methods subsection “Diet-induced obesity”. Briefly, animals were on low-fat chow from birth and tested on this diet at the outset of experiments. This is “baseline”. They were then randomized to either be switched to HFD for six weeks or maintained on chow for six weeks. After this six week period, the HFD cohort was returned to chow for four weeks and the chow cohort was maintained on chow for an additional four weeks.

This is important given there is considerable variability in the baseline groups in several panels possibly contributing to the perceived changes (e.g. Figure 1G-I; Figure 3; Figure 4E-G; Figure 5A-C; Figure 5E-G; Figure 4—figure supplement 1A-D). What accounts for this variability? The significance or lack of significance for Figures 4G, 5C, and possibly others appear to have benefited from this variability.

We have summarized below our interpretation of any variability between baseline groups in the various figures.

In optogenetic experiments in Figure 3, the differences in food intake between the baseline (no stimulation) groups in each panel represent real biological differences due to the different conditions tested. For example, fasted mice (Figure 3D, E) eat more than fed mice (Figure 3B, C) and mice presented with a HFD (Figure 3C, E) eat more than mice presented with a low fat diet (Figure 3B, D).

In the photometry experiments in Figures 1, 4, and 5, there is inherent variability in the magnitude of the responses of animals to stimuli at baseline due to the nature of photometry recordings. For example, small differences in fiber placement and viral expression cause differences in baseline fluorescence, and this technical variation cannot be prevented. The variability between animals that we report here is typical of these kinds of studies and the effect sizes we report (e.g. 30-40% DF/F) are larger than most photometry experiments in the published literature. However, we show the individual data points for each mouse in every experiment so that the reader can independently evaluate the spread in the data.

Several additional observations argue that variability in baseline responses is not a major contributor to the difference we observe. For example, we show that some signals are reversibly altered by weight loss (e.g. Figure 1E) and others are not (e.g. Figure 5B). Differences in the reversibility of a response cannot be explained by differential variability at baseline. Similarly, in some cases where DIO decreases AgRP neuron responses, the baseline responses were greater in the DIO group (Figure 5A-C), whereas in other cases where DIO decreases AgRP neuron responses, baseline responses were smaller in the DIO group (Figure 5G-I). Thus, differences in baseline responses do not reliably predict subsequent changes in dynamics.

Similar to the previous point; statistical comparisons should be made within and across groups at different times to assess these changes.

Statistical comparison between groups (lean and DIO) revealed no significant differences in AgRP neuron dynamics at baseline. The only significant between group comparison across all time points was the 6 week time points for chow and HFD presentation (Figure 1). This has been added to the figure. We have included information about relevant non-significant, between-group comparisons in the figure legends.

Overall, the manuscript is somewhat incremental. An exception is the fact that some of the attenuated responses to nutrients/food cues are long lasting after reintroduction of LFD for few weeks. A pair-fed control group would be critical to investigate the extent to which is the diet, and not the obesity itself, that causes the phenotypes described. Quantification of fat mass (and not only body weight) and other metabolic parameters related to DIO are important. The manuscript falls short in their analysis of the different components of DIO that can be contributing to the observed phenotypes.

The reviewer raises the question of what aspect of high-fat diet exposure is the underlying cause of the changes in AgRP neuron activity. We agree that this is an interesting question, but it is not the focus of this paper and, to be properly addressed, would require repeating the entire study. We do show that there is no correlation between weight gain and the change in the photometry response to presentation of either chow or HFD (Figure 1—figure supplement 1). We also show that is no correlation between weight gain and change in photometry response to CCK administration (new Figure 5—figure supplement 1I). This suggests that obesity and body weight per se do not drive the changes we observe in AgRP neuron dynamics in DIO animals.

In the revised manuscript, we include data from new experiments in which we subjected wild type animals to HFD (similar to our experimental animals) and then measured hormone levels and behavior and compared to lean controls (new Figure 2—figure supplement 1) and discussion beginning in the last paragraph of the subsection “Diet-induced obesity decreases food consumption after fasting”). These DIO mice showed significantly blunted fasting-induced refeeding and also developed hyperleptinemia. Importantly, circulating leptin levels negatively correlated with fasting-induced refeeding in DIO mice, suggesting that hyperleptinemia may be the cause of the blunted fasting induced hyperphagia in DIO mice, at least early in the course of obesity. In contrast, fasting insulin levels in DIO mice were not different from lean controls and do not correlate significantly with fasting induced refeeding.

References:

Chen, Y.M., and Knight, Z.A. (2016). Making sense of the sensory regulation of hunger neurons. Bioessays 38, 316-324.McFarlane, M.R., Brown, M.S., Goldstein, J.L., and Zhao, T.J. (2014). Induced ablation of ghrelin cells in adult mice does not decrease food intake, body weight, or response to high-fat diet. Cell Metab 20, 54-60.